# Biopolymer-Based Films Enriched with *Stevia rebaudiana* Used for the Development of Edible and Soluble Packaging

**Roxana Puscaselu *** **, Gheorghe Gutt and Sonia Amariei**

Department of Food Engineering, Stefan cel Mare University of Suceava, Suceava 720229, Romania;
g.gutt@fia.usv.ro (G.G.); gutts@fia.usv.ro (S.A.)
*  Correspondence: roxana.puscaselu@fia.usv.ro

**Abstract:** Currently, there is an increasing concern toward the plastic pollution of the environment, in general, and of oceans, in particular, as a result of disposable packaging in the food industry. Thus, it is extremely necessary that we identify solutions for this problem. This study was aimed at identifying a viable alternative—biopolymer-based, edible, and renewable food packaging—and succeeded in doing so. For this work, 30 films with different characteristics and properties were obtained using agar and sodium alginate as film-forming materials and glycerol for plasticization. Tests were performed, such as physical properties, microstructure, mechanical properties, microbiological characteristics, and solubility assessment, showing that edible materials can be used to package powdered products and dehydrated vegetables, or to cover fruits and vegetables, cheese slices, and sausages. These materials come from renewable resources, are easily obtained, and can be immediately applied in the food industry, thus being a viable alternative to food packaging.

**Keywords:** agar; sodium alginate; food; ecosystem; hydrocolloids

## 1. Introduction

Taking into account the fact that the total amount of plastic use currently exceeds 200 million tons annually, with an increase of about 5% per year, it is essential that alternative materials be used instead of conventional, non-recyclable and/or non-biodegradable packaging, thus solving many ecological problems. A relevant example is the Great Pacific Garbage Patch—in the central North Pacific, where plastic waste has formed an artificial island of about one million $km^2$, the result of marine pollution [1]. At present, tons of plastic packaging are discarded into nature, making the waste problem a bigger one year after year. Recycling the packaging would be impractical and economically inconvenient, because of the residues from food products and other biological substances. Under the circumstances, the use of biodegradable and eco-friendly biopolymer-based packaging represents the best candidate for reducing the huge amount of conventional plastics, being "bioplastics, manufactured exclusively with natural polymers, which are able to be fully recycled into the environment, into a short time, following their utilization as packaging materials" [2].

Biopolymers are defined as polymeric materials produced from renewable raw materials or biodegradable fossil fuels [3]. Bioplastic materials are a biopolymer subgroup, being entirely produced from bio or renewable resources that may or may not be degradable or compostable [4,5]. Not all bioplastics are compostable in the domestic environment, like organic food waste, and they usually require a special industrial composting treatment [6]. Without proper labeling and efficient collecting schemes, bioplastics can contaminate recycling streams, reducing the amount of recycled conventional plastics [7,8]. Therefore, it is a matter of the utmost urgency to find an alternative to synthetic and

conventional food packaging and the impossibility of its total disintegration. Therefore, we have concluded that the use of polysaccharide biopolymers may meet the above conditions in order to develop a biofilm intended for the packaging of powdered products, such as cappuccino, soluble coffee, powdered milk, and other types of powdered products, and dehydrated concentrated vegetables, which require solubilization prior to consumption.

Steviol glycozides (from *Stevia rebaudiana*) are non-caloric sweeteners used in the food and beverage industry, approved by the European Commission through Regulations 1333/2008 on food additives and 1334/2008 on flavorings. In the USA, stevia leaf and crude extracts are not considered GRAS and do not have FDA approval for use in food, even if the people of Japan have widely used stevia for decades [9]. Due to its chemical composition, it is considered to be the best sugar substitute, especially for patients with diabetes [10–12]. The benefits of stevia consumption are mainly due to its biochemical composition but also to its nutritional value; it is a good source of carbohydrates, proteins, fibers, minerals, amino acids, sterols, chlorophyll, organic acids, and inorganic salts [13,14]. The most important bioactive compounds are alkaloids, flavonoids, tannins, and phenolic compounds that are able to improve and prevent many diseases [15–18]. Thus, edible films and coatings often containing antimicrobial agents and/or other food additives, including anti-browning agents, colorants, flavors, nutrients, and spices, are gaining relevance as potential tools to reduce the deleterious effects of the incidence of microorganisms that can endanger the health of the consumer and change the qualities of the products [19].

Due to these benefits, stevia is a sugar substitute. The purpose of this research was to obtain edible films intended for the packaging of instant drinks, where sugar from the packaging contents could be substituted by stevia from its composition. In this way, if the consumer did not want to ingest the stevia, a non-sweetened product could be obtained by simply disposing of the biodegradable packaging. In addition, these edible films can be eaten by adults, children, elderly people, or individuals with special needs.

## 2. Materials and Methods

### 2.1. Materials

The biofilms were obtained from agar, sodium alginate, glycerol, stevia, and water. For the production of biofilms made from agar, sodium alginate, and glycerol, the addition of stevia reached 1.25% (Table 1). Except for agar (which was made available by B & V. The Agar Company, Parma, Italy), the sodium alginate and glycerol were purchased from Sigma-Aldrich (Romania Order Center, Bucharest, Romania). Stevia (Rebpure RA97) was purchased from a local distributer of GLG Life Tech Corporation (Richmond, BC, Canada). For the preparation of a 150 mL solution, the chemical amounts used in the trials were as follows: 0–3.45 g agar, 0–3.45 g sodium alginate, 0.50–1.50 g glycerol, and 0.05 g stevia powder.

**Table 1.** Ingredients and related quantities used to obtain films.

| SAMPLE | $m_{AGAR}$, (g) | $m_{ALGINATE}$, (g) | $m_{STEVIA}$, (g) | $m_{GLYCEROL}$, (g) | $V_{APĂ}$, (mL) |
|--------|--------|----------|---------|-----------|--------|
| S1 | 2.95 | 0.00 | | 1.00 | |
| S2 | 0.00 | 2.95 | | 1.00 | |
| S3 | 1.00 | 1.95 | | 1.00 | |
| S4 | 1.95 | 1.00 | | 1.00 | |
| S5 | 2.00 | 0.95 | 0.05 | 1.00 | 150.00 |
| S6 | 0.95 | 2.00 | | 1.00 | |
| S7 | 0.50 | 2.45 | | 1.00 | |
| S8 | 2.45 | 0.50 | | 1.00 | |
| S9 | 1.25 | 1.70 | | 1.00 | |

**Table 1.** *Cont.*

| SAMPLE | $m_{AGAR}$, (g) | $m_{ALGINATE}$, (g) | $m_{STEVIA}$, (g) | $m_{GLYCEROL}$, (g) | $V_{APĂ}$, (mL) |
|---|---|---|---|---|---|
| S10 | 1.70 | 1.25 | | 1.00 | |
| S11 | 3.45 | 0.00 | | 0.50 | |
| S12 | 0.00 | 3.45 | | 0.50 | |
| S13 | 1.00 | 2.45 | | 0.50 | |
| S14 | 2.45 | 1.00 | | 0.50 | |
| S15 | 2.00 | 1.45 | | 0.50 | |
| S16 | 1.45 | 2.00 | | 0.50 | |
| S17 | 1.50 | 1.95 | | 0.50 | |
| S18 | 1.95 | 1.50 | | 0.50 | |
| S19 | 1.25 | 2.20 | | 0.50 | |
| S20 | 2.20 | 1.25 | 0.05 | 0.50 | 150.00 |
| S21 | 1.00 | 2.20 | | 0.75 | |
| S22 | 2.20 | 1.00 | | 0.75 | |
| S23 | 1.50 | 1.70 | | 0.75 | |
| S24 | 1.70 | 1.50 | | 0.75 | |
| S25 | 2.00 | 1.20 | | 0.75 | |
| S26 | 1.20 | 2.00 | | 0.75 | |
| S27 | 1.25 | 1.95 | | 0.75 | |
| S28 | 1.95 | 1.25 | | 0.75 | |
| S29 | 1.60 | 1.60 | | 0.75 | |
| S30 | 1.225 | 1.225 | | 1.50 | |

*2.2. Methods*

The membranes were obtained by casting method, using different proportions of biopolymers, plasticizers, and stevia, according to Table 1. The film-forming solution was kept at 90 °C for 30 min, being continuously stirred, then poured, leveled, and left to dry on a silicone surface at a temperature of 23 °C (48 h approximately). Once obtained, the films were preserved under well-established temperature and humidity conditions (22 °C and 42% RH) and tested for safe use as packaging material.

2.2.1. Determination of Physical and Optical Properties

In order to evaluate the physical characteristics of each membrane, a number of determinations were made, such as color, taste, or smell, adhesion to the drying surface, thickness, and retraction ratio. The color was evaluated by the CieLAB system using a Chroma Meter CR400 colorimeter (Konica Minolta, Tokyo, Japan) and was established after at least five readings taken in different biofilm areas. The thickness was measured with an electronic digital micrometer (Mitutoyo, Kawasaki, Japan) with a precision of about 0.001 mm, measurements carried out in at least five random locations, and the average thickness value noted.

Retraction ratio was calculated using Equation (1), where the initial film thickness represents the film-forming solution thickness [20]. For this purpose, a silicone frame was used to determine the exact thickness of the initial solution (772.00 μm).

$$\text{Retraction ratio, } (\%) = \frac{\text{initial film thickness} - \text{dry film thickness}}{\text{initial film thickness}} \times 100\% \tag{1}$$

Film transmission was identified with an Ocean Optics HR 4000 CG-UV-NIR spectrometer (Ocean Optics, Douglas, AZ, USA) at 660 nm wavelength. An important property of the material intended for packaging which may be affected by light is its transmittance. On the supermarket shelf, products are often degraded due to light penetrating the packaging. Identifying this problem leads to finding viable solutions that prevent this degrading process.

### 2.2.2. Evaluation of Mechanical Properties

In order to determine the mechanical properties, the samples were tested for tensile strength and elongation at break by using an ESM 301 - Mark 10 texturometer (Stefan cel Mare University, Suceava, Romania) and the grips for thin films and films (Addex Design, Sibiu, Romania) as attachments. For determination purposes, STAS ASTM D882-02 (Standard Test Method for Tensile Properties of Thin Plastic Sheeting) was used [21]. Tests were performed at ambient temperature of 24.4 °C. The load of the machine was 5 kN and the speed was 10 mm/min. Three replicates of strips were cut at dimensions of 100 mm × 10 mm. The results of tensile strength were noted according to Equation (2), where *F* represents the maximum force supported by the sample and *S* represents the biofilm surface:

$$TS, \ (\text{MPa}) = \frac{F}{S}. \tag{2}$$

Elongation on break, also known as fracture strain or tensile elongation on break represents the ratio between increased length ($\Delta l$) and the initial length ($l$) [21]; it is the ability of plastic to resist changes of shape without cracking. The elongation is calculated as a relative increased in length, as per Equation (3):

$$\text{Elongation} \ = \frac{\Delta l}{l} \times 100\% \tag{3}$$

The determination of the roughness and microstructure of the membranes was performed with a MarSurf CWM100 microscope (Mahr, Gottingen, Germany), the results being noted after observing at least seven different areas. Mountain Map®software (Version 7, Digital Surf, Lavoisier, France)—surface imaging, analysis, and metrology software—was used in order to analyze the microscopic structure and reflectivity of the film surface.

### 2.2.3. Evaluation of Microbiological Characteristics

Whenever food or another product is intended for ingestion, it must be safe for consumption, that is microbiologically safe. Thus, both the obtained films and the ingredients used were tested for the identification of total counts, coliforms, enterobacteria, *Escherichia coli*, *Staphylococcus aureus*, yeasts, and molds. For this purpose, Compact Dry TC/CF/YM/XSA, EC dehydrated specific culture media (NISSUI Pharmaceutical, Tokyo, Japan) was used. A number of thermostatic conditions were maintained for the determination of each microorganism. To perform the determinations, 1 g of the film was solubilized in 9 mL of physiological saline; for each sample, three dilutions were made in order to clearly establish the total count. Subsequently, 1 mL of the obtained solution was poured onto the specific culture medium, and then let dry at 37 °C for 26 h (for total count, coli forms, enterobacteria, *E. coli*, and *S. aureus*), and for 72 h for yeasts and molds.

### 2.2.4. Solubility Assessment

Material solubility is an important parameter, since the material is intended as packaging for pulverulent products, completely dissolvable in hot water. To establish the suitable film for the development of such material, a series of determinations were made to characterize the material according to moisture content, water solubility, and water activity index. For moisture determination, film samples (3 cm × 3 cm) were weighed and maintained at 110 °C for 24 h [22]. They were then

reweighed, and the results were noted in the moisture calculation Equation (4), where W0 represents the initial sample weight and W1 the weight of the sample after drying [22]:

$$MC = \frac{W0 - W1}{W0} \times 100\%. \tag{4}$$

Testing the solubility in water implied the use of the same sample size; thus, 3 cm × 3 cm pieces were cut, weighed, immersed in 50 mL water for 8 h (22 °C temperature), dried in a hot air oven Memmert (Memmert, Schwabach, Germany) at 110 °C for 24 h, and reweighed [23]. For the films' water solubility characterization, the results were noted in Equation (5) [23]:

$$WS = \frac{W0 - W1}{W0} \times 100\%. \tag{5}$$

Even if Equation (5) is similar to humidity Equation (4), the functioning differs significantly.

The water activity index ($a_w$) was determined using AquaLab equipment (ICT International, Armidale, NSW 2350) at 22.8 ± 1.5 °C. The recorded value represents the sum of five determinations. In order to achieve the determinations, the samples were maintained under conditions of 42% RH and 20 ± 2 °C temperature.

All determinations presented in this section were performed in triplicate.

*2.3. Statistical Evaluation*

Statistical analysis was performed using Design Expert® Version 11 (trial version). Pearson correlation was made using SPSS (trial version).

## 3. Results and Discussion

*Characterization of the Obtained Materials*

The obtained films were soft, fine, thin, and flexible, without pores or cracks in the structure, intense shine, with no odor but a sweet taste, unlike other added powders, such as banana, which have influenced the development of films with a high degree of roughness and small particles on the film surface [24]. In addition, they showed low adhesion to the silicone support used for drying, had regular edges, and were pleasant to touch. Regarding physical characteristics, they can successfully compete with conventional packaging. Thickness was reduced; sample S30, with the highest glycerol content (1.50 g), showed a value of 37.80 ± 0.50 µm. The thickest film was S24 (52 ± 0.58 µm).

The retraction ratio is an important determination when the production recipe is used at the industrial level, but it also indicates the interactions between hydrocolloids during drying. Thus, the results indicate the possibility of controlling the final thickness of the membrane. For example, if a film with the ingredients and sample thickness S1 is desired and the value of the retraction ratio is known (35.78 ± 0.68 µm in this case), then the manufacturer must pour the film-forming solution and level it so that it has an initial thickness of approximately 73.15 µm. Color evaluations showed that edible films have no significant effect on *L** and *a** values: the minimum brightness value *L** was 93.02 ± 0.28 for S11 and the maximum was 95.84 ± 0.54 for S2. Similarly, the minimum value of the parameter *a** was −7.34 ± 0.28 (S2) and the maximum value was −6.24 ± 0.21 (S19), and for *b** the minimum was set to 18.58 ± 0.30 (S2) and the maximum was set to 20.95 ± 0.77 (S16 and S19). Small variations are due to the fact that the biopolymers used, agar and sodium alginate, are almost similar in color.

The complete solubility of samples S2, S12, S26, and S27 determines the ability of sodium alginate to produce biofilm that is completely dissolvable. Thus, reweighing the samples was not possible, hence the lack of values in Figure 1. The results indicate a material with good characteristics if we take into account that the beet-added films dissolved after about 3 min in water at temperatures above

90 °C [25] and even the fruit powder in the form of the tablet dissolved completely after 7 min in water [26].

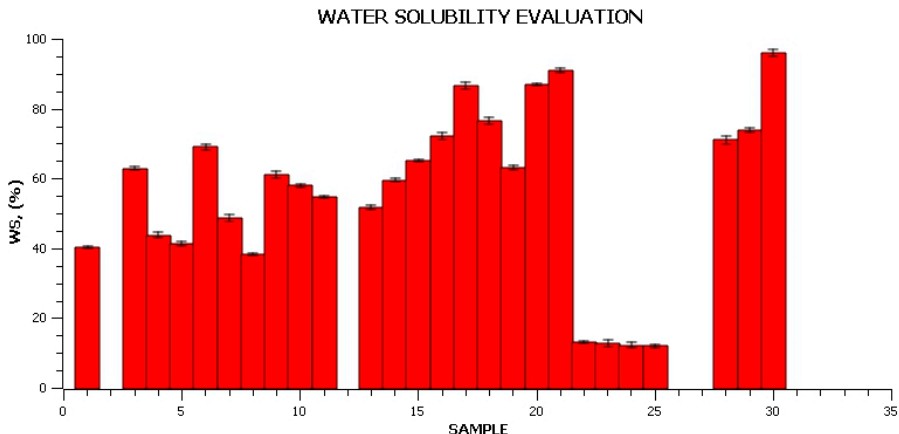

**Figure 1.** Water solubility of the films with addition of stevia powder.

The other samples with agar in the composition retained their integrity even after 20 min immersion in water at ambient temperature (21 ± 2 °C). According to the obtained results, we can conclude that sodium alginate has the ability to form films with higher solubility than agar to prevent water absorption, although biofilms obtained from equal or relatively equal amounts of biopolymers (S17, S18, S29, and S30) showed high solubility. The packaging material is intended for pulverulent products that require solubilization in water with temperatures above 80 °C. This material can also be used for sliced sausages or cheeses when there is a higher amount of agar and less sodium alginate in the composition. Similarly, it can also be used in the pharmaceutical industry (Figure 2), where stevia-added films can be an alternative to pharmaceuticals, frequently sweetened with artificial sweeteners (saccharin, cyclamate, aspartame, Acesulfame K) [27].

Following Tables 1 and 2, it is possible to establish a correlation between the composition of the films and the mechanical characteristics. For example, samples S1, S14, and S15 with high agar content in the composition and less sodium alginate exhibit high breaking point, high breaking strength, and superior elasticity. Films with a high content of sodium alginate in the matrix exhibit high elasticity (S2, S3, S7, and S9). The ability of sodium alginate to create homogeneous films together with agar is observed, as opposed to mixing it with chitosan, where films with reduced mechanical properties have been obtained [28].

Table 2 presents the Pearson correlation matrix between physical and mechanical characteristics of tested biofilms. It seems that some parameters (thickness, retraction ratio, tensile strength, elongation) are highly positively influenced one by the other ($p < 0.05$). Another consideration is that thickness is negatively correlated with transmittance ($Tr$; $r = -0.600^*$, $p < 0.05$) and brightness ($L^*$; $r = -0.657^*$, $p < 0.05$). Furthermore, we can observe a negative correlation between roughness and transmittance or brightness.

Microstructure, homogeneity, uniformity, and presence of pores or cracks are noted in Table 3. We must note that pores are found on the membranes' surface and that the material is not perforated in any of the cases (the maximum depth is 421.00 ± 0.16 nm).

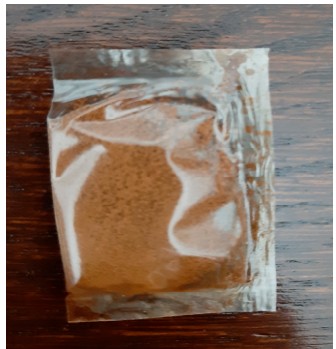 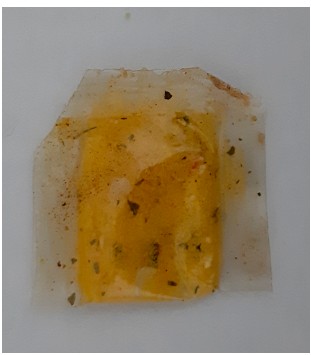 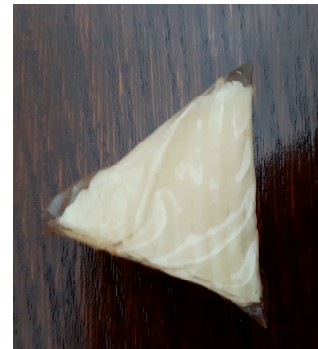 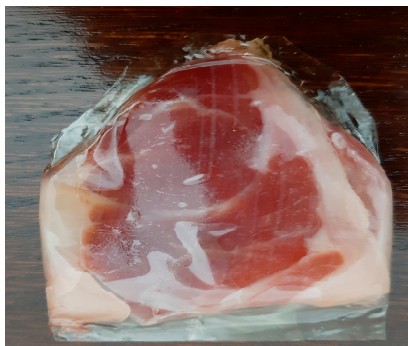 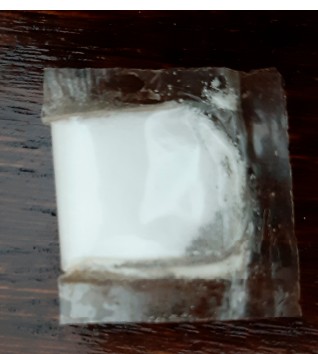

**Figure 2.** Applications of edible biofilms incorporated with *Stevia rebaudiana* (soluble coffee, dehydrated vegetables, cheese slices, meat slices, and medicine).

**Table 2.** Pearson correlation matrix between physical and mechanical parameters.

| | *T* | *Rr* | *TS* | *E* | *$R_z$* | *Tr* | *L\** | *a\** | *b\** |
|---|---|---|---|---|---|---|---|---|---|
| *T* | 1 | 0.905$^{(**)}$ | 0.850$^{(**)}$ | 0.853(**) | 0.351 | −0.600$^{(*)}$ | −0.657$^{(*)}$ | 0.067 | 0.589$^{(*)}$ |
| *Rr* | | 1 | 0.999$^{(**)}$ | 0.343 | −0.245 | 0.127 | 0.013 | −0.352 | 0.066 |
| *TS* | | | 1 | 0.340 | −0.249 | 0.132 | 0.127 | −0.354 | 0.077 |
| *E* | | | | 1 | −0.265 | 0.267 | 0.234 | −0.102 | −0.250 |
| *$R_z$* | | | | | 1 | −0.347 | −0.317 | 0.101 | 0.358 |
| *Tr* | | | | | | 1 | 0.743$^{(*)}$ | −0.081 | −0.322 |
| *L\** | | | | | | | 1 | −0.245 | −0.751$^{(**)}$ |
| *a\** | | | | | | | | 1 | −0.016 |
| *b\** | | | | | | | | | 1 |

*T*: thickness; *Rr*: retraction ratio; *TS*: tensile strength; *E*: elongation; *$R_z$*: roughness; *Tr*: transmittance; *L\**: brightness; *a\**: red-green axis; *b\**: yellow-blue axis; * Correlation is significant at the 0.05 level; ** Correlation is significant at the 0.01 level.

**Table 3.** Images and microstructures of materials suitable for use as food packaging.

| Sample | Appearance | Microstructure |
|---|---|---|
| **S2** With very small pores; no cracks, high homogeneity and full solubility of the biopolymers. | | |
| **S17** Homogeneous membrane, without pores, cracks, and non-solubilized ingredients. | | |
| **S22** High porosity area (pore diameter 8.52–13.3 nm and depth 142–305 nm) and irregular appearance. Negative particles are deposited on the film structure. | | |
| **S24** Membrane with superior characteristics; although it has asperities on the surface, the structure is more homogeneous. | | |

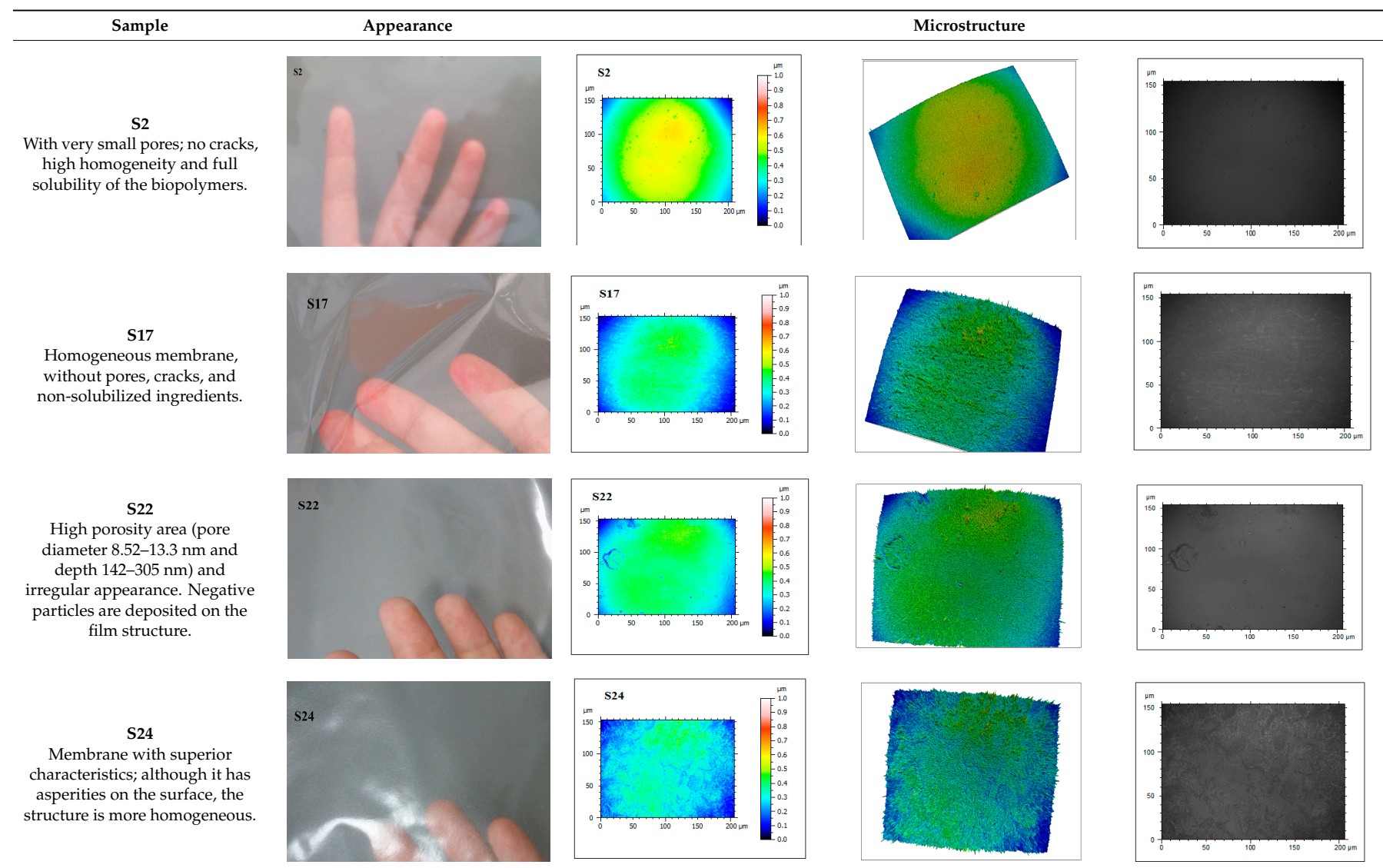

Figure 3 shows the obtained optimization results using a statistical investigation program (Design Expert 11). It should be noted that stevia's mass is constant (0.05 g), and the thickness and the retraction ratio are assumed to be 50.00 μm and 30%, respectively.

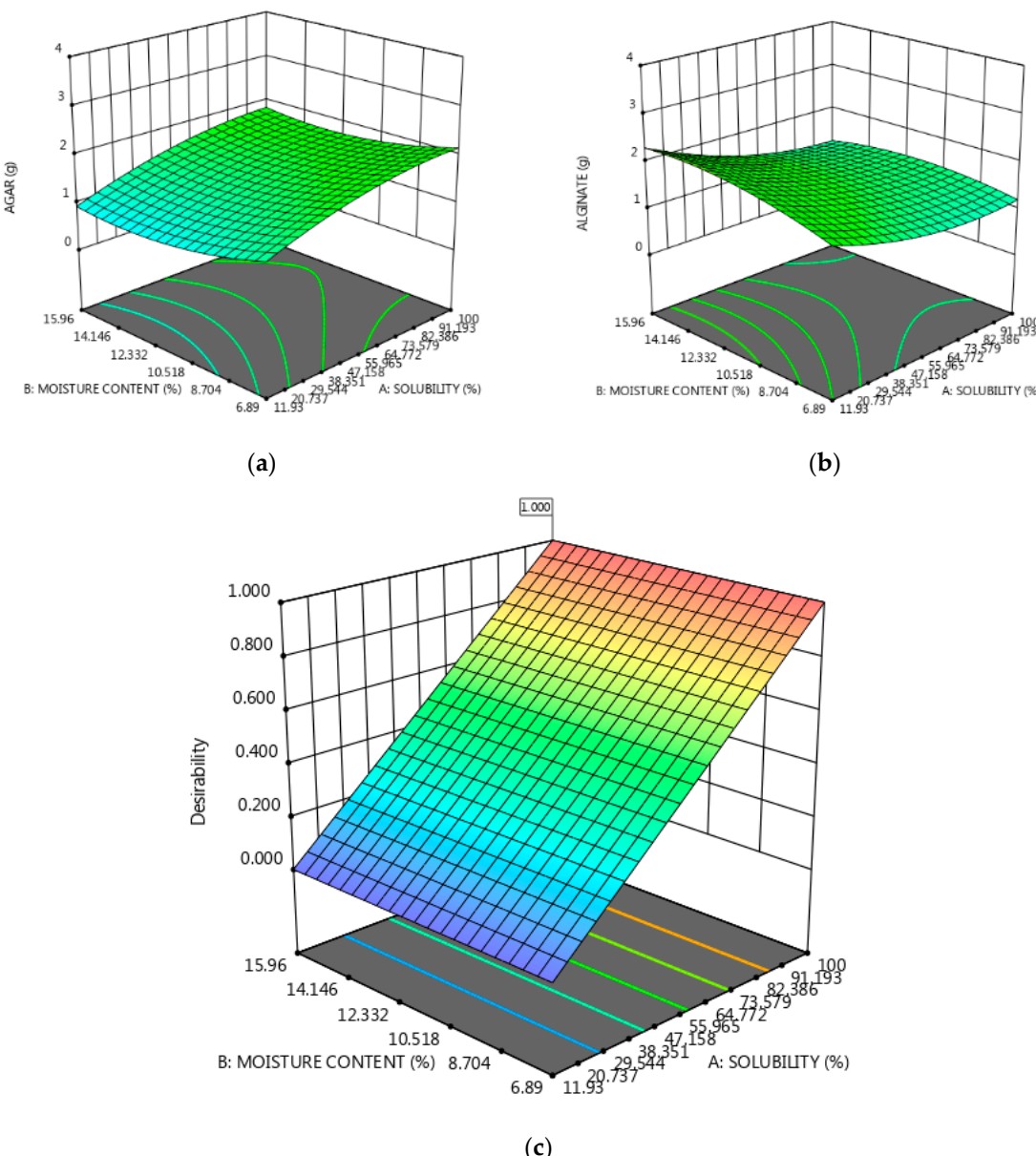

(a)

(b)

(c)

**Figure 3.** Optimizing the quantities used to create a completely soluble material. (**a**) agar content (**b**) alginate content; (**c**) desirability.

The optimization for the development of a material with superior mechanical and physical properties - breaking strength of 3.00 MPa, elongation of 84.80%, and roughness of 122.20 nm - was achieved using the recipe provided in Figure 4, with amounts of 1.634 g agar, 1.305 g alginate, and 0.919 g glycerol, respectively. We must note that the stevia content does not change (constant mass of 0.050 g). It should also be noted that the glycerol mass is more than half the amount of agar, with glycerol favoring the appearance of soft and smooth films having better mechanical properties, unlike other plasticizers (sorbitol, xylitol, mannitol) [29].

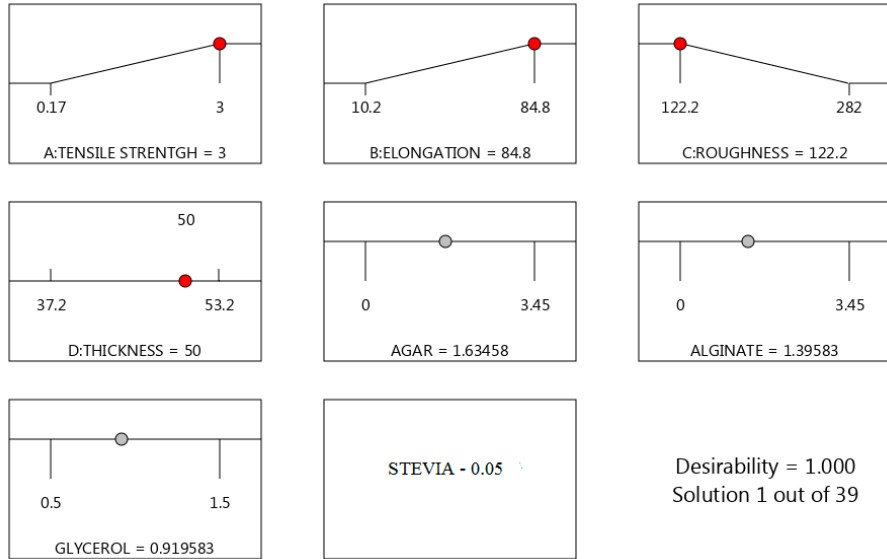

**Figure 4.** Ingredients and related quantities to obtain a material with good mechanical properties.

The moisture content was directly proportional to the amount of sodium alginate in the composition—S2, S3, and S19 (Figure 5).

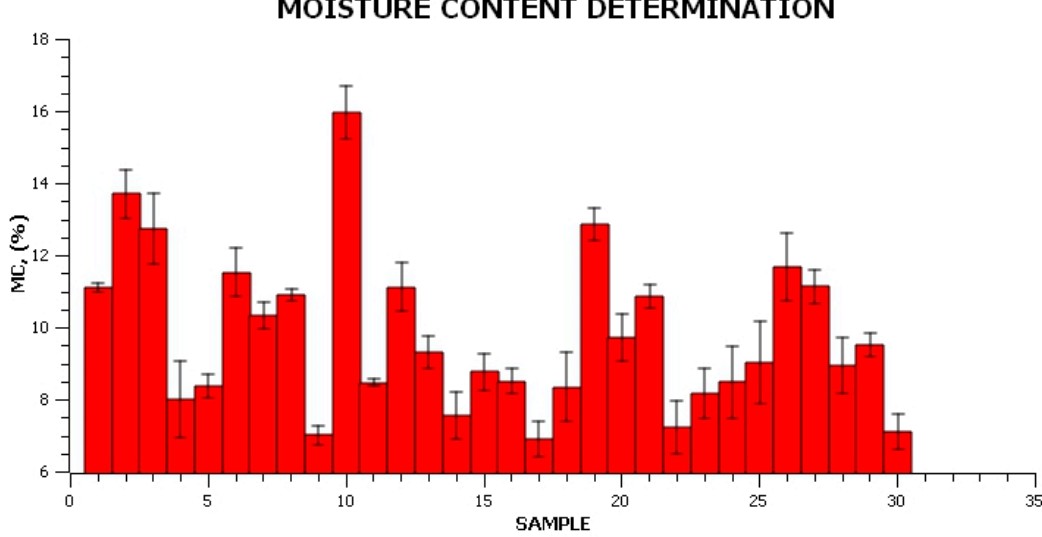

**Figure 5.** Moisture content evaluation.

However, the maximum moisture content of sample S10 can be observed, obtained from approximately equal amounts of agar, sodium alginate, and glycerol.

Microbiological determinations indicated the safe use of these membranes, the results showing the absence of tested microorganisms—coliform bacteria, enterobacteria, *E. coli*, *S. aureus*, yeasts, and molds. Both the biofilms and the ingredients are safe for consumption. The same behavior was revealed in the films obtained from starch and the addition of blackberry pulp, whose water activity index was 0.6 and which has shown inhibitory activity toward *S. aureus*, *E. coli*, or other coliforms [30].

The subsequent development of microorganisms is unlikely, especially because the films have low water activity index values (Figure 6), which do not favor the optimal conditions necessary for the incidence or proliferation of microorganisms.

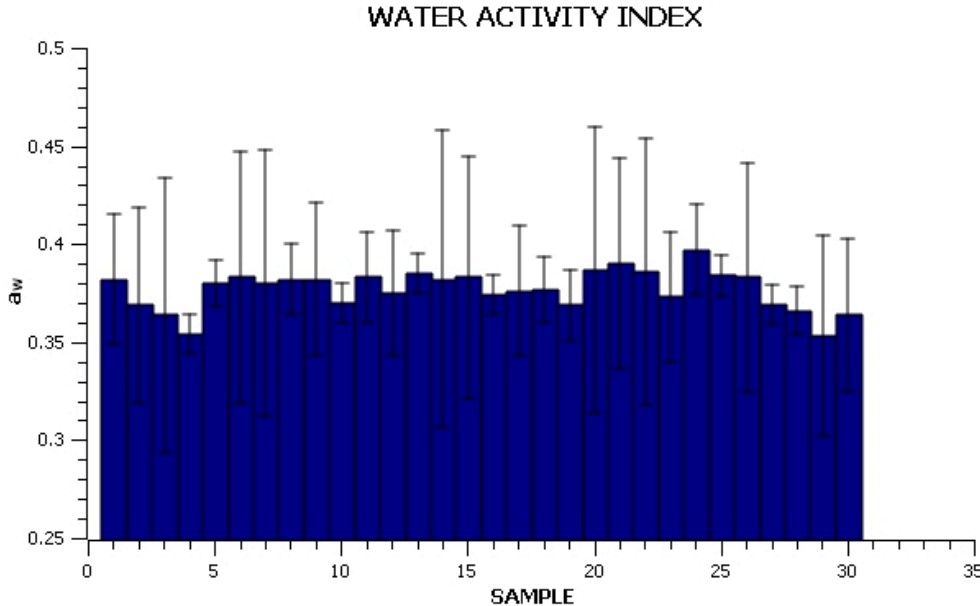

**Figure 6.** Water activity index of tested films.

Since this packaging is edible, it must not pose any microbiological danger to the consumers, especially when they are children or individuals with special needs. There are studies in the literature that indicate the safety of these films. Moreover, they can be enriched with added essential oils [31].

## 4. Conclusions

Edible films and coatings, made from natural ingredients, have become a viable alternative to oil-based packaging material. The study aimed at designing packaging materials for powdered products such as soluble coffee, cappuccino, powdered milk, teas, and dehydrated vegetables, with the possibility of extending them to the pharmaceutical field.

The tests indicate the use of polymers (agar, sodium alginate) and glycerol to obtain food packaging materials. The addition of *Stevia rebaudiana* leads to thinner, more flexible, and shinier materials. At the same time, solubility has increased, a desired aspect in this case. The presented recipes and the indicated quantities result in the production of edible packaging materials that can replace synthetic conventional packaging. The material obtained only from alginate and stevia (S2) has optimal characteristics for the use of powder-type packaging: high solubility, homogeneity, regular margins, medium roughness, good strength, and elasticity. For high humidity products or those with photo-degradable compounds, it is possible to control this aspect by using a relatively equal content of hydrocolloids (S24: 1.70 g agar, 1.50 g sodium alginate, and 0.75 g plasticizer).

The optimal recipe for a material with superior mechanical properties and minimal roughness is identified. The consumption of such biofilms is safe, as demonstrated by microbiological tests. No further occurrence of microorganisms is likely, as indicated in the extremely low values of the water activity index. Even when stevia is added in small quantities, it becomes an important ingredient of materials that can be used successfully in the food industry and other related industries. The material thus obtained can be easily reproduced in the industrial environment, it does not require special equipment, and the costs are greatly reduced due to its simple technique of production, but especially because of the renewable nature of the raw materials.

**Author Contributions:** Conceptualization, R.P., S.A. and G.G.; methodology, R.P. and S.A.; software, R.P.; validation, R.P., S.A. and G.G.; formal analysis, R.P.; investigation, S.A.; resources, G.G.; data curation, R.P.; writing—original draft preparation, R.P.; writing—review and editing, S.A. and G.G.; visualization, R.P.; supervision, S.A. and G.G.; project administration, G.G.; funding acquisition, G.G.

**Funding:** This work was supported under Contract No. 18PFE/16.10.2018 funded by the Ministry of Research and Innovation within Program 1—Development of national research and development system, Subprogram 1.2—Institutional Performance—RDI excellence funding projects.

**Conflicts of Interest:** The authors declare no conflict of interest.

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
