# Peer review of "Biopolymer-Based Films Enriched with Stevia rebaudiana Used for the Development of Edible and Soluble Packaging"

_coatings, doi:10.3390/coatings9060360_

Round 1

Reviewer 1 Report

Dear Authors,

Please evaluate/consider the following points, which were stated for the intention of  improving the quality of your submitted manuscript.

I wish you every success in your next projects.

1)     (Line 2) please check again if the designed coating can be considered as „intelligent packaging“?

Intelligent packaging materials and Intelligent Packaging Systems are defined by European Commission as; “materials and articles which monitor the condition of packaged food or the environment surrounding the food; Intelligent packaging systems provide the user with information on the conditions of the food”.

Reference : Web page https://ec.europa.eu/food/sites/food/files/safety/docs/cs_fcm_legis_active-intelligent_guidance.pdf

2)     (Line 13) The sentence indicates that agar, sodium alginate and glycerol (all three) are used as plasticizer. However agar and sodium alginate are film-forming materials. Please add an explanation in the sentence for clarification.

3)     (Line 16-17) „can be used for the packaging of .... cheese slices (not slices cheese).

4)     (Line 18) would you please check the term „profile industry“?

5)     (Line 25) Please check the reference style. [2,3] instead of [2], [3].

6)     (Line 29-30) Please check the sentence.

7)     (Line 35- 37) Please add the legal status of stevia as a food additive in EU/U.S.,….

8)     (Line 43 -45) It is unclear that if the consumers do not want to digest/consume sweet products, will Stevia not be incorporated in the formulation?   

9)     (Line 51-53) Please check the Materials & Methods Protocols of the journal. The manufacturer of an equipment/chemical can be written as e.g. X material (company name, city, country).

10)   (Line 52-53) Please write the details of Local Manufacturer of Stevia.

11)  (Line 50-53) Please include the amounts of chemicals (min and max values) used in the trials. E.g. agar (0.00g-3.45g), alginate (0.00g-3.45g) ... for 150 ml solution preparation. This can give a quick, clear idea to the readers without the necessity of checking the numbers in Table 1.

12)  Table 1. Is there any particular reason why these concentrations were chosen? If these concentrations were chosen according to previous studies, these studies should be cited.

13)  (Line 59) approximately

14)  (Line 59-60) Please specify the „well-established temperature and humidity conditions“.

15)  (Line 63) Please add informations about how „taste/smell“, „adhesion to the drying surface“, „thickness“, „retraction ratio“ tests were conducted with specifying the details of sensory tests, devices, etc. In addition please give the equation, used for calculation of retraction ratio.

16)  (Materials and Methods section) Please write the number of replication (n=...) conducted for each test (As it is stated in solubility assessment test line 96).

17)  (Line 80-83) Please write the specific growth media that were used and incubation conditions & time for each test microorganism.

18)  (Line 90-93) Please give the equation that was used for calculation of water solubility (WS) amount. In addition, although it was stated that the experiment was conducted according to the study of Wang and Rhim (2015), the drying conditions (i.e. temperature and time) used in the present study are slightly different than the study of Wang and Rhim (2015). Please state the modifications shortly.

19)  (Line 91) Please check „same type of sample“. I believe that it should be „same size of sample/film“.

20)  (Material & Methods) Please add a „Statistical Evaluations“ subsection.

21)  (Line 99) Table 2, 3 and 4 were referred in text before figure 1. Please give the tables, figures, etc. directly after the first time they were referred.

22)  (Line 103) please write the average numbers of the results of the measurements with standard deviations throughout the manuscript.

23)  (Line 104-108) Please check the results of retraction ratio evaluations. Retraction ratio is not only used in industrial applications but also indicates the interactions between hydrocolloids during drying. Therefore I recommend to give the results of the retraction ratio test.

24)  (Line 109) Please evaluate your data with statistically and add the p-value to „parameters varied extremely little“.

25)  (Line 118, Figure 1) Please add standard deviations to the graph.

26)  (Line 120) According to Figure 1, S28 dissolves also completely in water. Therefore please also add S28 in the text.

27)  (Line 124-127) In an experimental design conducted in the present manuscript, the effect of agar, alginate and glycerol should be evaluated statistically to be able to make this conclusion. For example, Three-way ANOVA and a post hoc test can show that if the factors have significant effect on WS and which levels are significantly different than others. Please evaluate the data first before coming to this conclusion.

28)  (Line 128-131) As stated previously, please conduct statistical evaluations to determine the significance of the effects.

29)  (Line 133, Table 2) Please state the samples, which are significantly different. You can also consider to present them in graphs. Evaluation of the results in such a long table, is very hard for the reader.

30)  (Line 135, Table 3) Please indicate the statistically different results.

31)  (Line 138, Table 4) Please check the explanations for S17.

32)  (Line 144) Design-Expert is a statistical software package that specifically performs design of experiments (DOE). Please mention about the program in Material & Methods section in “Statistical Evaluations” subsection.

However, I would like to underline that it is still necessary to determine the statistically significant effects in each test (Line 98-131).

33)  (Line 153, Figure 4) Please add standard deviations.

34)  (Line 169-171) Please consider if you can make this conclusion at this point. In the present manuscript, No data of conventional/synthetic packaging was presented. Moreover, not any actual comparison between synthetic packaging and designed-edible-films were presented.

Author Response

Dear Reviewer,

First of all, I would like to thank You for your comments, they have been really useful for me, taking into account the status of PhD student and the constant accumulation and necessity of new information.

Secondly, I want to answer the points received.

You were right about "intelligent packaging" taking into account the EC definition. I would like to estate the novelty and the characteristics of edible packaging, define them intelligent. I didn't know that really exist regulatory on this issue. We removed the word from the title of the paper.

I clarified the sentence.

, 5, 6, 9, 11, 13, 19, 22, 31 - i made the suggested revisions

when i wrote "profile industry" i meant, first of all, at food industry, but this coatings can be use in medicine or pharmacology industry, as well.

 7. I added the legal status of stevia as food additive - Regulations 1333/2008 on food additives and 1334/2008 on flavourings. In US stevia leaf and crude extract doesn't have the GRAS and FDA approvals, even in Japan have widely used stevia for decades,

8. I would have wanted to explain that the consumption can be simply refused by throwing the packaging. It's not like stevia is found in the composition of the product and you can't refuze the consumption.

10. Stevia was purchased from a company from an ice cream Romanian factory that use stevia in their products. I wrote into the manuscript this aspect, too.

12. I choose these concentrations based on my laboratory research. After several attemps, i decided to use these concentrations.

14. 42 % RH and 20-22 degrees.

15. I noted all the formulas.

20, 24, 27, 28, 29, 30, 32. I added Pearson matrix to estate the correlations between results and determinations. a note a statistical evaluation subsection, as well.

16, 19. I noted these information.

17. For microbiological evaluation, we used Compact Dry plates with lyophilized  culture medium (Japan). Thus, each plate contain specific culture medium, but all contains peptone and speicif yeasts. The incubation conditions and time: 37 degrees/24 hours for bacterias and 37 degrees/72 hours for yeasts and molds (it is a unique plate for both type of microorganisms).

34. I would like to estate the benefits in use of there edible packaging material.

Thank You!

Reviewer 2 Report

This manuscript is quite interesting. Anyway there are many thinks that should be fixed up. 

-First of all please provide statistical analysis.

-please describe in a more exhaustive way themicrobiological analysis. 

-please discuss th significance of the microbiological analysis

-in the introduction please review the followig papers:G. Rossi Marquez, et al.  Fresh-cut fruit and vegetable coatings by transglutaminase-crosslinked whey protein/pectin edible films. LWT - Food Science and Technology. 75: 124–130, 2017; Giosafatto et al. (2019) Effect of mesoporous silica nanoparticles on glycerol-plasticized anionic and cationic polysaccharide edible films. CoatingsVolume 9, Issue 3, 2019, Article number 172.

Please extend the introduction.

Author Response

Dear Reviewer,

First of all, I would like to thank You for received comments which are very useful for me, as a PhD student.

I extended the introduction and i reviewed the suggested papers; there are really interesting, with a great work and conclusions. I created a Pearson matrix with my results in order to highlight the correlations between determinations and results obtained. I put standard errors on graphics, as well.

Regarding mircobiological analysis, I think this is an important aspect if we take into account that this packaging is edible and must be safe for the consumer. For determinations, we used Compact Dry plates (Japan), which contains specific lyophilized culture media. Growth conditions are same as classical method: 37 degrees/24 h for bacteria and 37 degrees/72 h for yeasts and molds. For analysis, i took 1 g of film and added 9 ml of physiological serum. I mixed well and fast (in order to prevent film dissolution, took 1 ml of solution thus obtained and placed on plate). I made three dilutions, and no or developed microorganisms in any plate. 

Thank You!

Reviewer 3 Report

line 35 

 Stevia (Stevia Rebaudiana) is a natural sweetener used in the food and beverage industry. 

It is not correct! Steviol glycosides (from Stevia) are non-caloric sweeteners. 

line 36

 best substitute for sugar? in scientific literature it is necessary to white chemical compound (saccharose). 

line 50

stevia addition - as a powder???

line 52

Stevia was obtained from local certified manufacturers - specify! 

In a methods all devices need to have county of manufacturing.

line 138 

Table 4. Images and microstructures of materials suitable for use as food packaging. 

Appearance 

 need to be proved by spectrophotometry - not with the picture!

Packaging material suitable for 

??? Do you have any preliminary research on food and beverages? If not please delete that column - that is only a presumption that has not been proven!

line 143 

Figure 2. Optimizing the quantities used to create a completely soluble material. 

Values/numbers on a picture of solubility are not readable!

Results have to be statistically processed (Anova) and interpreted. Because like that have no meaning scientifically. 

line 168 

Stevia Rebaudiana in to italic Stevia Rebaudiana 

Conclusions have to be rewritten and interpreted with results (statistically) correlation.

Author Response

Dear Reviewer,

First of all, I would like to thank You for the received comments and suggestions. There are really important for me, as a PhD student.

Secondly, I want to answer at your unclear points: stevia, added as a powder, was purchased from a a local ice cream factory that use it for diabetics products. They purchased it from a Canadian producer and I noted this on my paper. 

I put my results on a Pearson matrix in order to correlate the determinations. I put the standard error bars on my graphs, as well. I used DOE statistical evaluation in order to establish exact composition for obtaining desirable materials. I made researches on food, but there are not published yet, but I insert some images with our coated products. As my doctoral research, I added a lot of substances into film matrix - carob powder, inulin, or essential oils.

Thank You for suggestions!

Round 2

Reviewer 1 Report

Dear Authors,

Please evaluate/consider the following points, which were stated for the intention of improving the quality of your submitted manuscript.

I also recommend an exhaustive grammatical revision.

I wish you every success in your next projects.

1.      (Line 66-67) Please explain the sentence. It is not clear what is this value (1.25%). Is it dry basis? w/w, or v/w? I guess that it is the proportion of stevia amount to the total amount of other ingredients (agar+alginate+glycerol). But it is very unclear in the sentence.

2.      Comment from 1st review: (Table 1). Is there any particular reason why these concentrations were chosen? If these concentrations were chosen according to previous studies, these studies should be cited.

i.     Authors‘ answer : „I choose these concentrations based on my laboratory research. After several attemps, i decided to use these concentrations.“

ii.     Comment from 2nd review: I believe that researchers who will be interested in the present research and want to repeat (and later improve) the experiments presented in this manuscript, will need this information. Therefore, I believe that the preliminary research should be mentioned briefly to clarify why these amounts were chosen. 

3.      (Line 80-81) Please clarify how you conduct the „taste/smell“ and „adhesion to the dry surface“ tests. If these tests were not performed using a device or proper methodology, only by the researchers‘ personal observations. Please erase them.

4.      (Line 90) Please check the equation (retraction ratio)

5.      (in Materials and Methods section) Please add the units to the explanation of each equation.

6.      (in Materials and Methods section) Please cite the sources where you take the equations from.

7.      (Line 120-123) Please note the compact dry media you use for each tests. E.g. Compact Dry dehydrated specific culture media (Compact Dry TC/EC/CF/YM, company, city, country).

8.      Line (157-160) Please check if you can state this conclusion. Are these physical characteristics determined by the researchers? Or are they scientific results of sensorial/physical tests performed by ... number of panelists?

9.      Line (160-161) Please add explanations to the thickness evaluations. For example. In which circumstances thickness was reduced? Can the thickness reduction or the reason why S24 has the highest thickness be explained by any phenomena?

10.   (Line 162-167) Please check the retraction ratio results and the discussions of the result. I disagree with the statement that „retraction ratio results are used in industrial level and irrelevant for the present manuscript“. Please check the discussions of the previous studies in literature. For example; The, D. P., Debeaufort, F., Voilley, A., & Luu, D. (2009). Biopolymer interactions affect the functional properties of edible films based on agar, cassava starch and arabinoxylan blends. Journal of Food Engineering, 90(4), 548-558.

11.    (Line 168, Line 172) statements such as „color parameters varied extremely little“ or „are almost similar in color“ are not scientific explanations. Please write your statistical evaluation results (SPSS results). E.g. instead of „extremely little“ researchers can write, color evaluations showed that edible films had no significant effect on L*/a*,... values (p>0.05).

12.   (statistical evaluations) Please check the previous articles in the literature about statistical evaluations and how the results were presented in the texts. E.g. Statistical evaluations showed that sodium alginate/agar, etc. concentration had significant effect on film thickness (p value). The Pearson's correlation can be used to evaluate the relationship between more than two variables. It does not show "variation" among and between groups.

13.   (Line 182-185) glycerol is not a film forming material. It is plasticizer. One film forming material should be compared with other film forming material.

14.   (Line 189-190) Table 2 was conducted for thickness; retraction ratio; TS,  E, roughess, transmittance; L*, a*, b and gave the relationship between these values only. Could you please check how you „establish a correlation between the composition films and the mechanical characteristics”

15.   (Line 190-191) „For example, samples S1, S14, S15 with high agar content in the composition and without sodium aginate exhibit high breaking point, breaking strength and superior 191 elasticity“.

S14 (2.45g Agar + 1g Alginate) and S15(2g Agar+1.45g Alginate). However, the sentence above suggests that S14 and S15 did not contain any alginate.

16.   (Line 249-256) The paragraph is not a conclusion. It is an introduction.

17.   (Line 258-259). Please check if you can make this conclusion. In the present manuscript, stevia was incorporated into all samples . Therefore there was no comparison between „samples with stevia„ and „samples without stevia“ to be able to conclude that „The addition of Stevia Rebaudiana aid in obtaining thinner, more flexibile, and shinier“.

Author Response

Dear Reviewer of the paper,

First of all, I want to thank you for reading the paper and for the observations made.

Together with the other authors, we replied to your notations as follows:

- we have improved the level of English

1. Stevia used was dry basis and represent 1.25% represent the proportion of stevia amount to the total amount of other ingredients. I have noted in the text and I hope the sentence is clearer now.

2. The concentrations were chosen after many laboratory tests. The article is based on the determinations made for my doctoral thesis and has not been taken from other studies. That’s why it was my response to the first reviewer. I have noted the exact composition along with precise quantities to be reproduced by those who want to make such biofilms. These films described in the articles represented the best variants from all points of view: physical, optical properties, mechanical determinations, solubility.

3.The tests were based only by researchers observations, so, at your suggestion, we have removed them from the text.

4,5,6, 7I made the notations.

8. I deleted the phrase

9. The thickness of the films has changed depending on the ingredients used. S24 contains a higher amount of glycerol to offset the other films, hence reducing the thickness.

10. I deleted “irrelevant for this manuscript”. It is a great observation. Thank You.

11, 12 I made the suggested changes.

13. I deleted the word “glycerol” and the comparison will be made between alginate and agar.

14. As mentioned above, the work is based on the entire work of doctoral research. I chose here to present the best variants and results. Thus, the one who will reproduce the determinations, obtain the best results.

15. I deleted and replace with “lower alginate”.

16. I deleted the paragraph.

17. We have noted these conclusions, because in the lab we have made films without the addition of stevie, that is why we consider the conclusion is relevant.

18. We did not want to clutter the article with all the attempts. But in my articles already published, various films made from agar, alginate, sometimes starch, enriched by the addition of essential oils, or carob powder. In the review process is another article, with the addition of inulin.

Respectfully,

Roxana PUSCASELU

Reviewer 2 Report

The manuscript has been extensively I.

Author Response

Dear article Reviewer,

Please allow me, above all, to thank you for reading our work and for the observations made.

We have improved the level of English.

Respectfully,

Roxana PUSCASELU

Reviewer 3 Report

 New Trends in Edible Food Packaging: the title is not needed.

Author Response

Dear article Reviewer, Thank you for reading the work and for the observations made. At your suggestion, I changed the title of the paper. Thus, it became “Biopolymer based films enriched with Stevia Rebaudiana, used for the development for edible and soluble packaging”. Respectfully, Roxana PUSCASELU